ⓐ | **Open Peer Review** | Environmental Microbiology | Research Article

# Altitude as a key environmental factor shaping microbial communities of tea green leafhoppers (*Matsumurasca onukii*)

Yong Zhang,[1,2,3] Song Liu,[1] Xue-yu Huang,[1] Hua-bin Zi,[1] Tian Gao,[1] Rui-jie Ji,[1] Juan Sheng,[1] Dian Zhi,[1] Ying-lao Zhang,[3] Chun-mei Gong,[2] Yun-qiu Yang[1]

**ABSTRACT** The tea green leafhopper, *Matsumurasca onukii* Matsuda, is the most destructive insect pest of tea plantations in East Asia. While several microbes in *M. onukii* have been characterized, the microbial community compositions in wild *M. onukii* populations and the environmental factors that shape them are mostly unknown. In this study, *M. onukii* populations were collected from major tea growing regions in China. Following high-throughput sequencing of 16S rRNA gene fragments for bacteria and the internal transcribed spacer region for fungi, association analyses were performed within the microbial communities associated with *M. onukii* and their environmental drivers. We found that the bacterial community structures differed in various regions, and the abundance of dominant bacteria such as *Wolbachia*, *Pseudomonas*, *Acinetobacter*, *Pantoea*, *Enterobacter,* and *Methylobacterium* varied widely. Moreover, wild populations of *M. onukii* can be infected with facultative symbionts from six genera (*Wolbachia*, *Rickettsia*, *Asaia*, *Serratia*, *Arsenophonus,* and *Cardinium*) with divergent relative abundances. Correlation analysis indicated that altitude was a key environmental factor that shaped bacterial communities of *M. onukii*. Furthermore, longitude, temperature, and rainfall are also significantly correlated with the bacterial communities. The fungal communities of *M. onukii* populations were dominated by Ascomycota and Basidiomycota, of which most genera are considered to be plant endophytes or plant pathogens, such as *Cladosporium*, *Fusarium*, *Alternaria,* and *Gibberella*. We demonstrated that *M. onukii* carry a complex and variable microbial community, which is influenced by altitude as well as climate-related factors. Our results provide novel insights into the bacteria and fungi of *M. onukii*.

**IMPORTANCE** Host-associated microbial communities play an important role in the fitness of insect hosts. However, the factors shaping microbial communities in wild populations, including environmental factors and interactions among microbial species, remain largely unknown. The tea green leafhopper has a wide geographical distribution and is highly adaptable, providing a suitable model for studying the effect of ecological drivers on microbiomes. This is the first large-scale culture-independent study investigating the microbial communities of *M. onukii* sampled from different locations. Altitude as a key environmental factor may have shaped microbial communities of *M. onukii* by affecting the relative abundance of endosymbionts, especially *Wolbachia*. The results of this study, therefore, offer not only an in-depth view of the microbial diversity of this species but also an insight into the influence of environmental factors.

**KEYWORDS** *Matsumurasca onukii*, microbiomes, tea green leafhopper, high-through-put sequencing

M icrobial symbionts are especially ubiquitous in insects, existing in insect exoskeletons, guts, and even intracellularly, and are usually beneficial or even necessary

Address correspondence to Yun-qiu Yang, longyanhua@ahau.edu.cn, Chun-mei Gong, gcm228@nwafu.edu.cn, or Ying-lao Zhang, Zhangyl@ahau.edu.cn.

Yong Zhang, Song Liu, and Xue-yu Huang contributed equally to this article and share first authorship. The order of authorship accurately reflects our respective contributions to the study

The authors declare no conflict of interest.

See the funding table on p. 16.

for the survival of their insect hosts (1). Microorganisms can enable insects to thrive, and enhance their adaptation to various environmental changes (2). For example, microbiome infection frequencies determine the geographic distribution of the chestnut weevil, *Curculio sikkimensis* (3). It was found that higher tiers of endosymbionts (i.e., *Wolbachia*, *Sodalis,* and *Rickettsia*) were present in weevils at localities with higher temperatures; lower numbers of *Wolbachia* and *Rickettsia* were detected in a population found in regions with higher snowfall. Furthermore, microorganisms acquired from the surrounding environment can enhance the fitness of insect hosts (4, 5).

The tea plant [*Camellia sinensis* (L.) O. Kuntze] is an evergreen perennial that is widely planted in almost 30 countries such as China, Japan, India, Kenya, Indonesia, Vietnam, Sri Lanka, and Turkey, and it is one of the most important economic crops worldwide (6). China represents the largest tea producer, consumer, and exporter in the world. According to data from the Food and Agriculture Organization of the United Nations (http://www.fao.org), China has 3.12 million hectares of tea plantations and fields, producing 3.2 million tons of tea and exporting 369,400 tons of tea annually; in 2021, the output of the tea industry in China was more than US $2.29 billion. Tea plants in China are distributed over a wide geographical range, from 18°–37°N and 94°–122°E, including nearly 1,000 counties in 21 provinces and municipalities. In terms of vertical distribution, tea plantations in China are found at elevations varying from 4 to 2,600 m above sea level (7). The tea plantations in China are usually divided into mountain tea plantations (where the altitude is generally above 400 m, with a relative drop of more than 200 m) and hilly tea plantations (altitude <400 m) (8). In particular, Chinese tea plantations are characterized by complex terrain and a changeable environment. The young leaves and buds of tea plants are processed into a variety of teas, including green, white, yellow, oolong, black, and dark tea, which is a popular daily beverage (9). Tea production plays a role in the development of the agricultural economy in China; unfortunately, leaf sap sucking by leafhoppers can cause severe yield losses and a reduction in tea quality as a result of damage to the tea plants.

The tea green leafhopper, *Matsumurasca onukii* (formerly *Empoasca onukii*) (Hemiptera: Cicadellidae), is one of the most notorious insect pests of tea in the tea-growing regions of East Asia, causing up to 50% of economic losses in tea production annually in China and up to 33% in Japan (10, 11). Both nymph and adult *M. onukii* pierce and suck the sap of tender tea shoots, which are the most important part of the plant for producing high-quality tea (12). Adult females also lay their eggs in these shoots, leading to irreparable damage (13). *Matsumurasca onukii* can cause yield losses of between 15% and 50%, and up to 100% in severely damaged plantations (14). More importantly, *M. onukii* is well adapted to different tea varieties and geographical variations (15). Indeed, *M. onukii* occur in almost all tea plantations, and they are serious tea pests in most tea plantations. Little is known about why *M. onukii* is so well adapted to geographical variations, and to date, there has been no explanation from the perspective of the insect's microbiota. Recently, extensive evidence has shown that insect microbiota affect essential physiological functions in their insect hosts, including determining viable temperature ranges, modulating desiccation tolerance, supplementing essential nutrients, overcoming plant anti-herbivore defenses, broadening the range of suitable food plants, or strengthening of immune responses for protection against pathogens (3, 16).

To date, however, the microbial communities of *M. onukii* have not been elucidated. Whether *M. onukii* adaptability in relation to geographical variation is related to its microbial communities is unknown. Not only that, but the factors influencing microbial communities in wild populations, including environmental factors and interactions among microbial species, are still poorly understood. Given the tea green leafhopper's broad geographical distribution and high adaptability, it serves as a suitable model for studing the impact of ecological drivers on microbiomes. The aim of the present study was to fill this knowledge gap and provide a basis for a better understanding of the relationship between microbiota and environmental drivers. We sampled 20 *M. onukii*

wild populations from 14 geographically distinct provinces and municipalities, and tea plantations from two types of areas, namely hilly and mountainous locales, in mainland China. We performed thorough culture-independent high-throughput sequencing of the 16S rRNA gene for bacteria and the internal transcribed spacer (ITS) region for fungi and conducted a systematic investigation of the microbial communities associated with field populations of *M. onukii* that live in distinct environments. Our analysis demonstrated that *M. onukii* harbor a very diverse microbiota, which is influenced by altitude as well as climate-related factors.

## RESULTS

### Microbial community diversity

We first analyzed the bacterial microbes associated with wild *M. onukii* populations. Alpha diversity analyses showed that *M. onukii* harbor a highly diverse bacterial microbiota, with the Hilly samples having more diverse and abundant microbiota than the Mountain samples (overall median Shannon diversity 6.1 (range 2.6–7.1) versus 4.1 (range 1.4–6.8); overall median observed operational taxonomic units (OTUs) 353 (range 141–695) versus 230 (range 87–490), respectively; Kruskal-Wallis test; $P < 0.05$) (Fig. 1A and B). This was supported by the observation that 54.1% of OTUs were unique to the Hilly samples compared with 33.5% of OTUs that were unique to the Mountain ones (Fig. S4). The non-metric multidimensional scaling (NMDS) analysis using the distance based on weighted and unweighted UniFrac analyses showed that the bacterial community structures significantly differed qualitatively and quantitatively between Hilly and Mountain samples (weighted adonis: $R^2 = 0.16$, $P = 0.001$; unweighted adonis: $R^2 = 0.05$, $P = 0.001$), indicating that the bacterial communities varied among *M. onukii* populations (Fig. 1C and D).

Compared with those of bacteria, the fungal alpha diversity also differed significantly among wild *M. onukii* populations (Kruskal-Wallis test; $P < 0.05$). Fungal phylogenetic diversity and richness for Hilly samples were significantly lower than Mountain ones (overall median Shannon diversity 3.4 (range 2.1–5.4) versus 4.2 (range 3.4–6.4); overall median observed OTUs 371 (range 259–489) versus 588 (range 443–669), respectively) (Fig. 2A and B). The NMDS analysis using the distance based on unweighted UniFrac showed that the fungal community structures significantly differed between Hilly and Mountain samples ($R^2 = 0.11$, $P = 0.02$) (Fig. 2C). In contrast, the NMDS analysis using the distance based on weighted UniFrac showed that the fungal community structures did not differ significantly between Hilly and Mountain samples ($R^2 = 0.11$, $P = 0.11$) (Fig. 2D). This indicated that the dominant fungal compositions were similar between Hilly and Mountain samples. This was supported by the observation that 1,954 OTUs were shared by 2 groups and accounted for 67.5% and 75.9% of the total OTUs in Hilly and Mountain samples, respectively (Fig. S4).

### Microbial community composition

Phylogenetic analysis showed that the bacteria of Proteobacteria were the most abundant in high-abundance OTUs (i.e., in the top 10% in terms of relative abundance), followed by Bacteroidota and Firmicutes (Fig. S5). The dominant phyla (relative abundance >1%, defined by presence in at least 10% of samples) across all samples were Proteobacteria, which accounted by for >70%, followed by Bacteroidota, Firmicutes, and Actinobacteriota (Fig. 3A). Bacteria from these four phyla accounted for 95% of the identified reads in *M. onukii*. At the order level, the highest mean relative sequence abundance was Enterobacterales (25.1% ± 18.1%; mean ± SE), followed by Pseudomonadales (16.2% ± 16.6%), Rhizobiales (10.8% ± 9.1%), and Burkholderiales (8.5% ± 6.1%) (Fig. 4).

The relative abundance of the dominant bacterial genera differed significantly among *M. onukii* populations. The amplicon sequencing detected 30 dominant bacterial genera, including 5 endosymbiotic genera (i.e., *Arsenophonus*, *Wolbachia*, *Cardinium*, *Asaia,* and

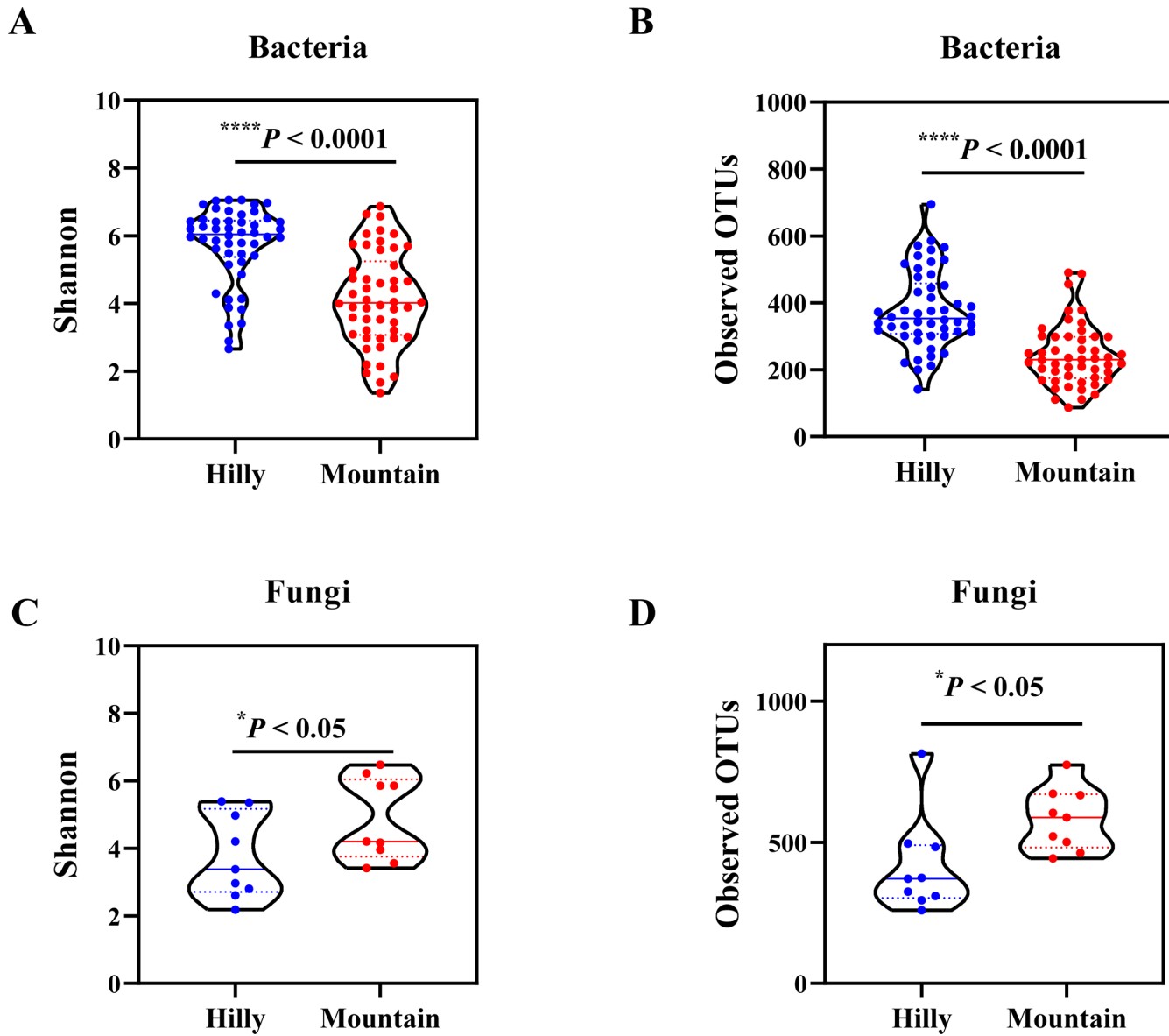

**FIG 1** Comparison of alpha-diversity of the bacterial and the fungal communities associated with *M. onukii*. Violin plots of (A and C) species diversity (Shannon index) and (B and D) species richness (observed OTUs). Hilly, samples collected from hilly area tea plantations; Mountain, samples collected from mountainous region tea plantations. The significant differences of alpha diversities were analyzed using the Kruskal-Wallis test and *P* values adjusted with false discovery rate method. Significance levels are denoted as follows: $^*P < 0.05$; $^{****}P < 0.0001$.

*Rickettsia*), and 25 ectosymbiotic genera of the identified reads in the *M. onukii* samples. Among those genera, three common endosymbiotic genera (i.e., *Wolbachia*, *Arsenophonus,* and *Rickettsia*) and seven ectosymbiotic genera (i.e., *Acinetobacter*, *Pantoea*, *Pseudomonas*, *Enterobacter*, *Methyloversatilis*, *Sphingomonas,* and *Rhizobium*) were dominant bacterial genera in at least 75% of the samples. Furthermore, *Pectobacterium* and *Microbacterium* were dominant genera in 45% of the samples (Fig. 3C). The linear discriminant analysis effect size (LEFSe) results, under the threshold of linear discriminant analysis (LDA) >4.0, showed that Mountain samples were rich in *Wolbachia*, *Pantoea,* and *Enterobacter*, while the Hilly samples were rich in *Acinetobacter* and *Methylobacterium* (Fig. 5).

Phylogenetic analysis revealed that Ascomycota were the most abundant fungi in high-abundance OTUs (i.e., the top 10% in terms of relative abundance), followed by Basidiomycota (Fig. S5). The dominant phyla across all samples were Ascomycota, which

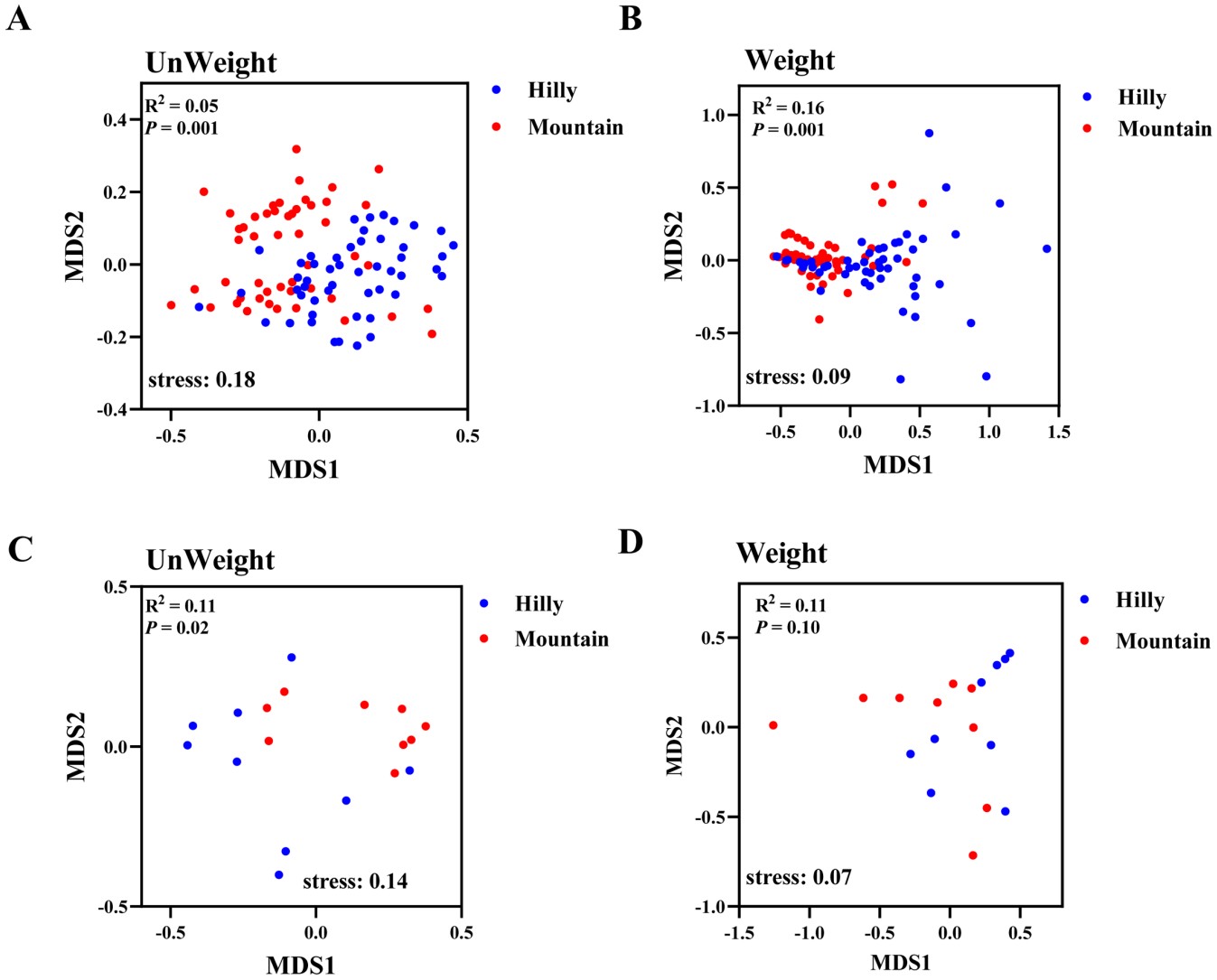

**FIG 2** Comparison of beta diversity of the bacterial and fungal communities associated with *M. onukii*. Unweighted (A and C) and weighted (B and D) UniFrac-based NMDS plots of bacterial and fungal communities. Hilly, samples collected from hilly area tea plantations; Mountain, samples collected from mountainous region tea plantations. The significant differences in beta diversities were analyzed using adonis analysis with 999 Monte Carlo permutations. Significance levels are denoted as follows: [*]$P < 0.05$; [***]$P < 0.001$.

accounted for >70%, followed by Basidiomycota (17.8% ± 16.9%) (Fig. 3B). Fungi from these two phyla accounted for 90% of the identified reads in *M. onukii*. At the order level, the dominant fungal order was Capnodiales (42.8% ± 16.6%), followed by Agaricales (13.6% ± 5.8%), Pleosporales (9.9% ± 4.6%), and Hypocreales (5.9% ± 1.6%) (Fig. 4). The amplicon sequencing detected 20 dominant fungal genera of the identified reads in *M. onukii*. Five common fungal genera (i.e., *Cladosporium*, *Psathyrella*, unidentified_*Pleosporales*_sp., *Penicillium*, and *Aspergillus*) were dominant fungi in at least nine test groups, which accounted for more than 50% of the total groups (Fig. 3D). There was no significant difference between the dominant fungal genera in Hilly and Mountain samples in the LEFSe (Fig. S6).

## Functional prediction of the microbial communities

Based on the Tax4Fun2 results, a total of 45 metabolic pathways (level 2) belonging to six Kyoto Encyclopedia of Genes and Genomes (KEGG) metabolic pathway groups (level 1) were predicted in all test samples. The KEGG pathway "Metabolism", accounting for up to

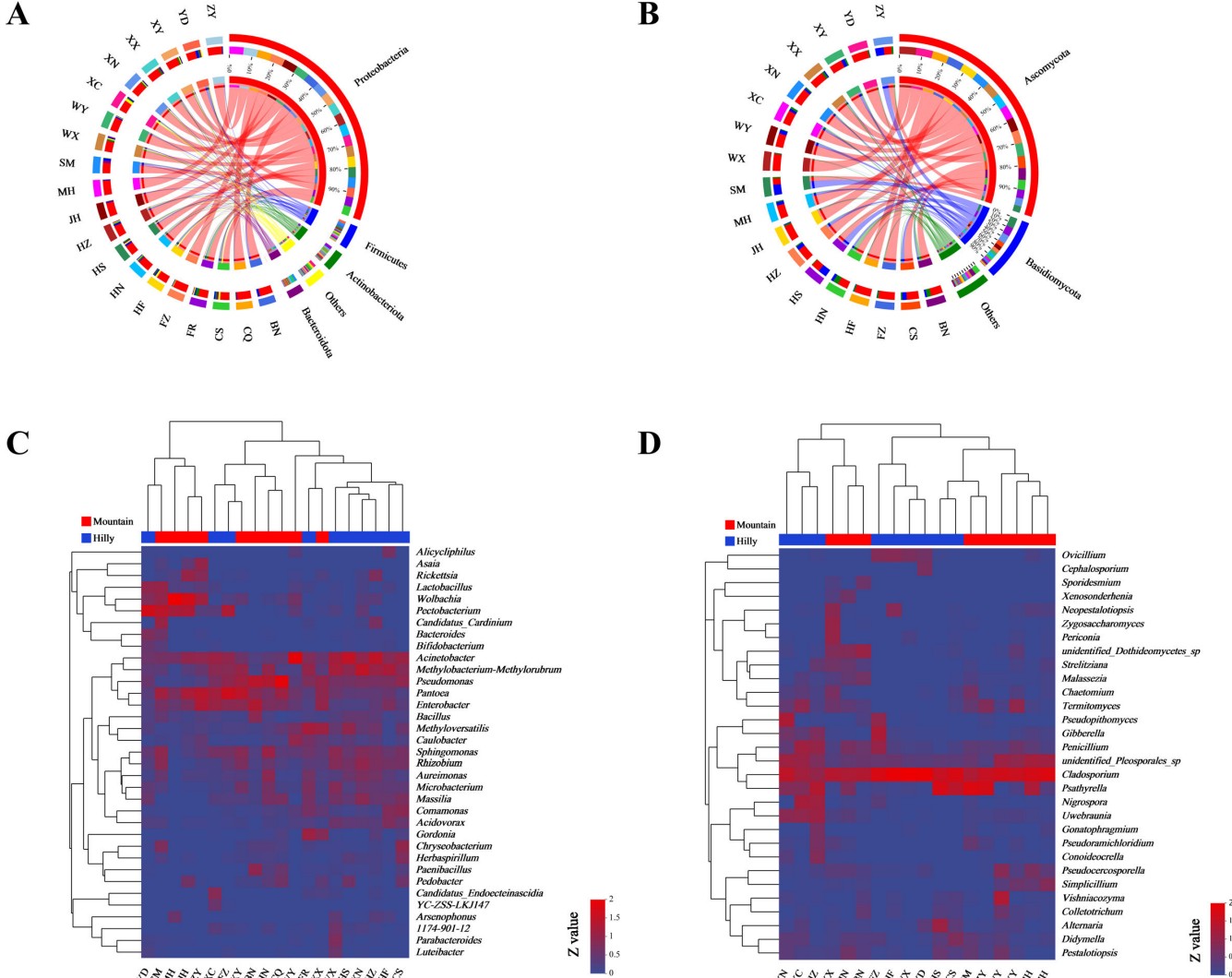

**FIG 3** Taxonmic composition of bacterial and fungal communities associated with *M. onukii*. Relative abundance of bacterial phylum (A) and fungal (B) phylum; each bar is indicated by a different color at phylum level. Heat-map analysis of the dominant bacterial genus (C) and fungal genus (D); Hilly, samples collected from hilly area tea plantations; Mountain, samples collected from mountainous region tea plantations. *Z* value = Log10$^{(relative abundance)\%}$ + 1; dominant microbial, relative abundance >1%, defined by presence in at least 10% of samples.

70% of the total relative abundance, was the most abundant metabolic pathway (Fig. 6A). These results indicated that the function of the bacterial communities associated with *M. onukii* were mainly to participate in the metabolism of a variety of compounds. In the metabolism category, the highest metabolic pathways in each group were global and overview maps (35.7% ± 1.2%; mean ± SE), followed by carbohydrate metabolism (9.1% ± 0.7%) and amino acid metabolism (6.9% ± 0.8%) (Fig. 6A; Fig. S7). Carbohydrates, amino acids and lipids are the three essential nutrients for animals. The heat-map results showed that the bacteria involved in the metabolism of carbohydrates, amino acids and lipids mainly came from Rhizobiales, Burkholderiales, Pseudomonadales and Enterobacterales (Fig. 6B).

FUNGuild classified three main trophic modes ("Pathotroph," "Saprotroph," and "Symbiotroph"), including eight functional guilds in our samples. In the assigned functional OTUs, Pathotroph-Symbiotroph and Saprotroph were the two highest trophic models in Hilly and Mountain samples, in which the relative abundance accounted for 73% and 56%, respectively (Fig. 7A). In the functional guild, the main trophic guilds in each group were Endophyte-Plant_Pathogen (37.9% ± 23.1%), Undefined_Saprotroph

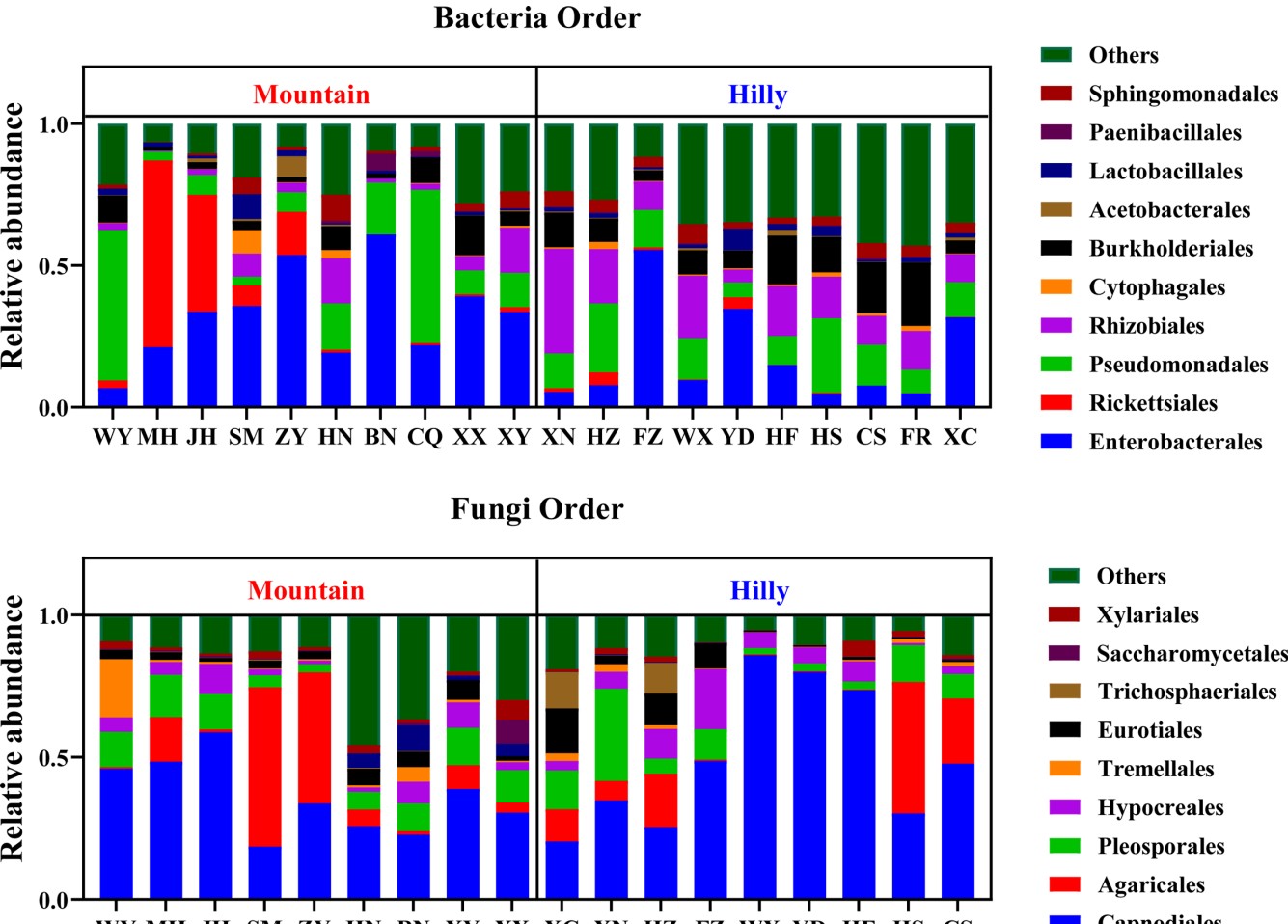

**FIG 4** Relative abundance of the *M. onukii* microbial community composition on order level. Each bar is indicated by a different color at order level; Hilly, samples collected from hilly area tea plantations; Mountain, samples collected from mountainous region tea plantations.

(14.1% ± 13.5%), Wood_Saprotroph (12.1% ± 17.7%), and Plant_Pathogen (5.6% ± 4.1%) (Fig. 7B).

## Bacterial community composition in *M. onukii* and their influencing factors

Because the dominant fungal communities had similar compositions, we only explored the correlation between environmental factors and bacterial communities. The canonical correspondence analysis (CCA) results suggested that two geographical factors (i.e., altitude, longitude) and three climate factors (i.e., Bio1, Bio12, TEM) were significantly correlated with bacterial community diversity and that altitude contributed the most toward explaining variation in bacterial community diversity (Fig. S8). The results of the Spearman correlation test revealed that altitude and Bio1 had a highly significant negative correlation with bacterial community alpha diversity, while longitude and TEM_Avg had a highly positive correlation with bacterial community alpha diversity ($P <$ 0.01) (Fig. 8A). These results were confirmed by the significant negative simple linear relationship shown via regression analyses, which were based on the Pearson correlation test (Fig. S9). The canonical principal coordinate (CAP) results suggested that altitude contributed the most toward explaining variation in bacterial community beta diversity for *M. onukii* ($P <$ 0.01, with significance determined by a permutation-based ANOVA test) (Fig. 8B; Table 1). Further analysis showed that altitude was significantly and positively correlated with the dominant bacteria *Wolbachia*, *Rickettsia*, and *Asaia* ($P <$ 0.05, $r >$ 0.4)

**TABLE 1** Constrained analysis of principal coordinates of the bacterial community compositions[a]

| Factors | Variance (%) | F | P |
|---|---|---|---|
| Altitude | 8.1 | 8.6 | 0.001 |
| Bio1 | 4.1 | 4.2 | 0.001 |
| Bio12 | 4.4 | 4.5 | 0.001 |
| Longitude | 7.6 | 8.1 | 0.001 |
| TEM_Avg | 4.6 | 4.8 | 0.001 |

[a]Bio1, annual mean temperatures; Bio12, annual mean precipitation; TEM_avg, the mean temperature for 30 days in total before and after the sampling date. Modules with *P*-values <0.05 are plotted.

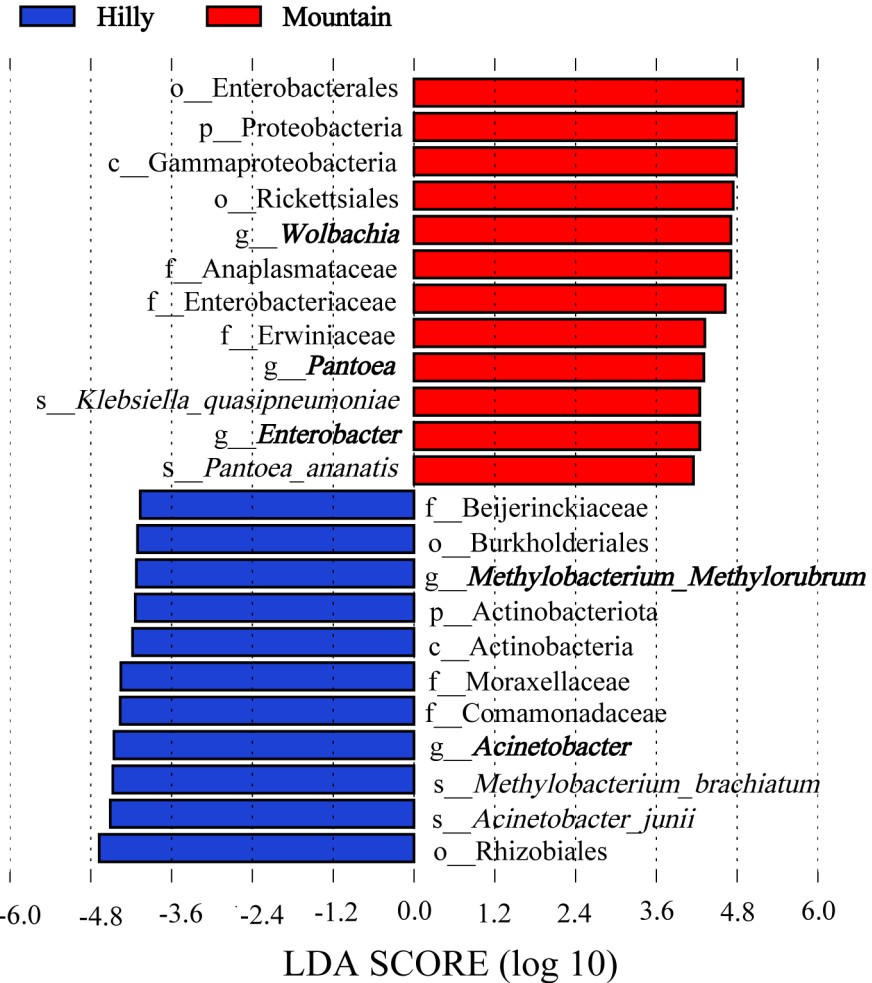

**FIG 5** Differential abundance of bacterial taxa associated with *M. onukii* as determined by linear discriminant analysis effect size. LDA score values represent the discriminatory power of features in a given data set. LDA scores >4 indicate that a particular feature has a stronger discriminatory power and can effectively distinguish between different groups; Hilly, samples collected from hilly area tea plantations; Mountain, samples collected from mountainous region tea plantations.

and significantly and negatively correlated with *Staphylococcus*, *Methylobacterium*, *Massilia*, *Microbacterium*, and *Rhizobium* (*P* < 0.05, *r* <−0.4) (Fig. 9).

## DISCUSSION

### Regional differences in bacterial communities of *M. onukii*

The microbiota of an organism is shaped by a complex set of factors (17). Currently, there is only limited research on the microbial community composition and diversity of *M.*

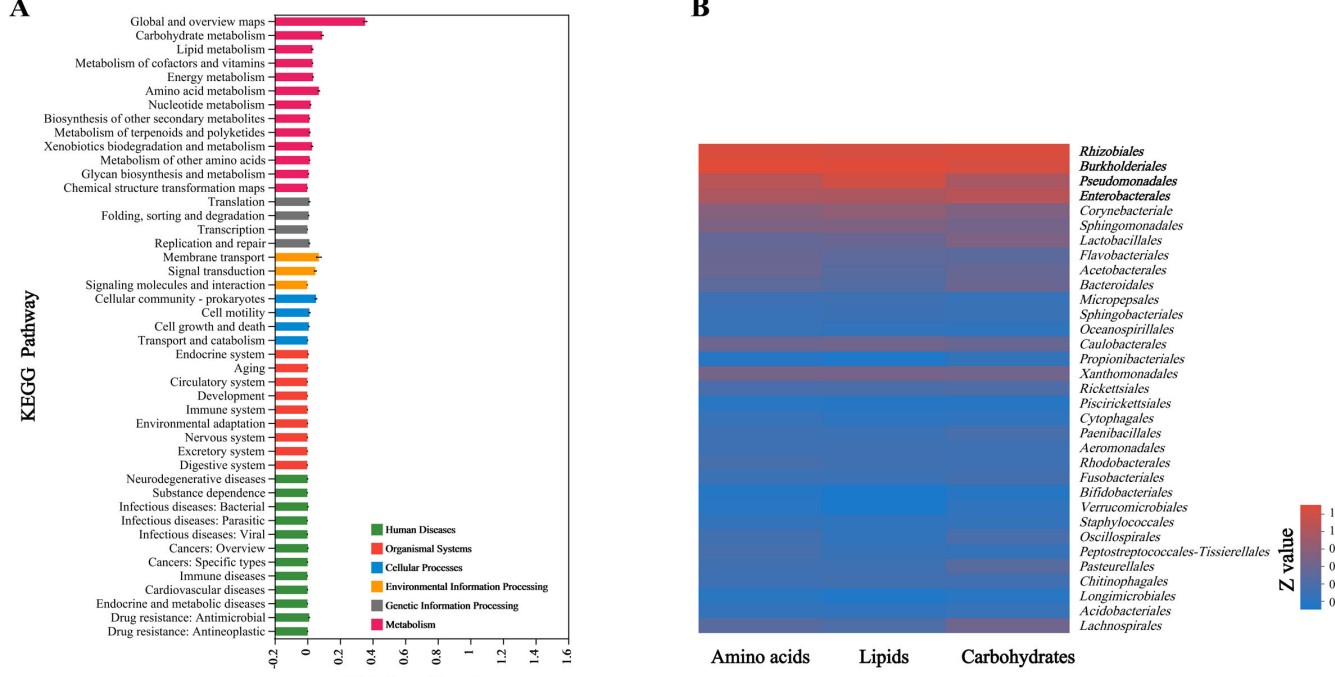

**FIG 6** Relative abundance of predicted functional bacterial community associated with *M. onukii* as determined by Tax4Fun2. The annotation results of the bacterial community of *M. onukii* at KEGG level 1 (A); heat-map analysis of the relative abundances (order-level) of bacteria involved in the metabolism of the three major nutrients (belonging to KEGG level 2) (B); $Z$ value = Log10$^{(relative\ abundance)\%}$ + 1.

*onukii*, as well as the key environmental factors affecting this species. To our knowledge, this study represents the first large-scale culture-independent investigation of the bacterial and fungal communities associated with *M. onukii* sampled at different locations. The results presented here not only further our understanding of the diversity of microbes in *M. onukii* and sap-feeding insects in general but also provide important insights into potential implications of environmental factors. On the basis of our collective sequencing evidence, we conclude that *M. onukii* harbor a very diverse microbiota, which is influenced by geographical as well as climate-related factors. Additionally, altitude as a key environmental factor may have shaped microbial communities of *M. onukii* by affecting the relative abundance of endosymbionts, especially *Wolbachia*.

In this study, *M. onukii* samples were collected from tea plants at 20 distinct geographic locations, including hilly and mountainous regions in China. Alpha diversity analysis showed that the bacterial communities of *M. onukii* collected from tea plantations in hilly areas were more abundant and diverse than *M. onukii* collected from plantations in mountainous regions. This finding is consistent with previous studies that demonstrated that altitude may contribute to bacterial diversity (18, 19). Not only that, the bacterial community structure of *M. onukii* collected from tea plantations in hilly areas was different from that of *M. onukii* collected from plantations in mountainous regions. This suggests that *M. onukii* harbor a highly diverse bacterial community and vary with geographical location.

At the phylum level, all samples showed high similarity, with Proteobacteria representing the dominant phylum at a relative abundance of over 70% followed by Firmicutes, Bacteroidota, and Actinobacteriota. This finding is consistent with the previous research conducted with other sap-feeding insects, such as aphids (*Sitobion miscanthi* and *Acyrthosiphon pisum*) and rice planthoppers (*Laodelphax striatellus* and *Sogatella furcifera*) (20–22). At the genus level, the predominant bacterial genera differed significantly among populations of *M. onukii*, such as *Wolbachia*, *Pseudomonas*, *Acinetobacter*,

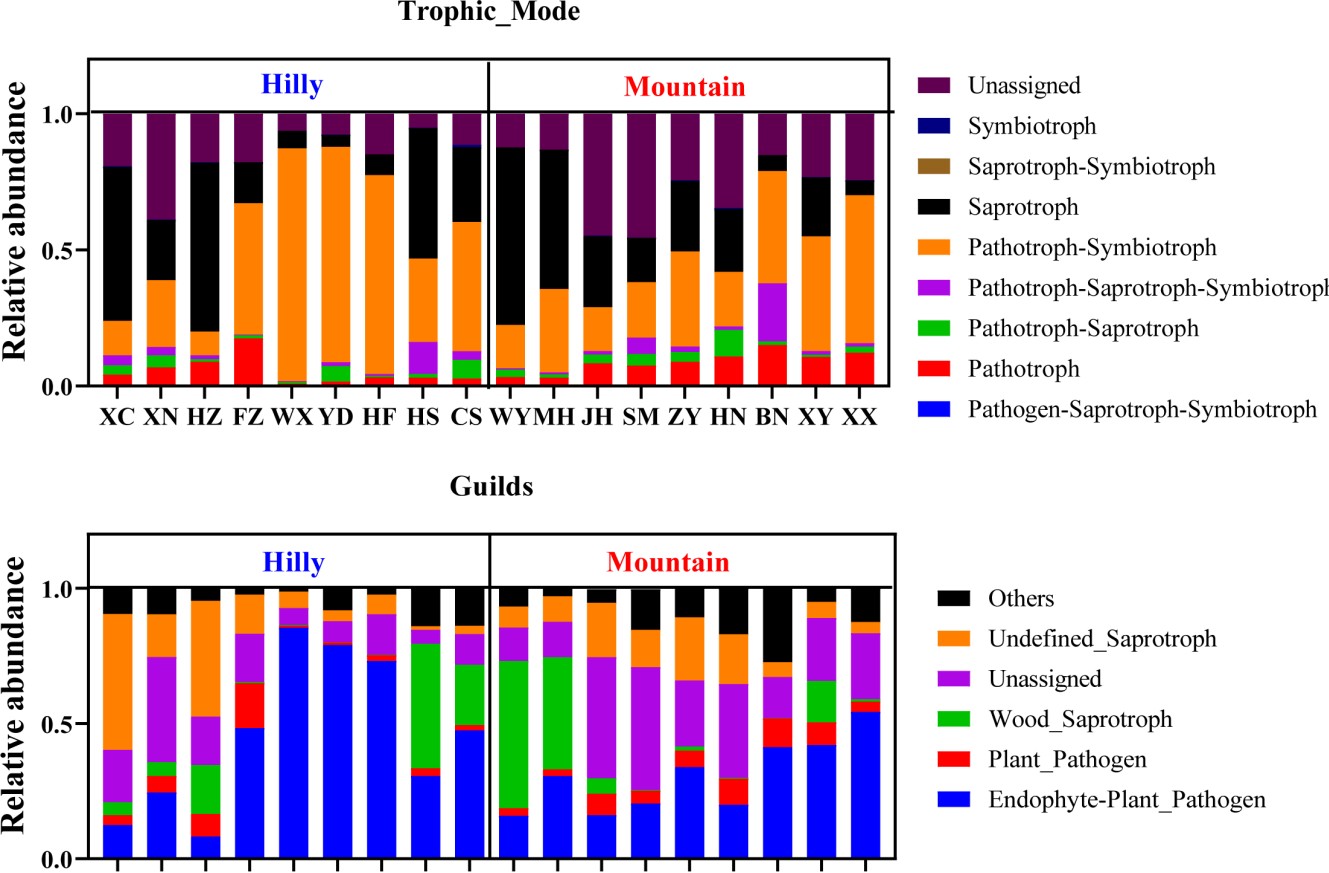

**FIG 7** Relative abundance of predicted functional fungal community associated with *M. onukii* as determined by FUNGuild. Trophic modes, the nutrient acquisition strategies employed by fungi; guilds, the functional guild classifications provided by the FUNGuild database. Hilly, samples collected from hilly area tea plantations; Mountain, samples collected from mountainous region tea plantations.

*Pantoea*, *Enterobacter,* and *Methylobacterium*. Wild populations of *M. onukii* can be infected with facultative symbionts from six genera (*Wolbachia*, *Rickettsia*, *Asaia*, *Serratia*, *Arsenophonus,* and *Cardinium*) with divergent relative abundances. Despite the potential functional importance of these symbionts, they were not detected in some of the *M. onukii* populations collected from different geographic regions, which indicated that the persistence of a symbiont is dependent on the balance between cost and benefit. In addition to endosymbiotic bacteria, extracellular bacteria (mainly gut bacteria) *Acinetobacter*, *Pantoea*, *Pseudomonas*, *Enterobacter*, *Methyloversatilis,* and *Sphingomonas* were more widely distributed and had higher infection rates, which could mean that these bacteria play a specific role in host growth and development, and they should be further studied.

Hemipterans that are phytophagous usually feed on nutritionally deficient xylem or phloem diets, but bacterial communities in these insects can provide essential amino acids and other nutrients (23, 24). The Tax4Fun2 analysis results showed that bacterial communities in *M. onukii* were mainly involved in nutrient metabolism. Interestingly, the bacteria involved in the three essential nutrients (carbohydrate, amino acids, and lipids) were mainly from Rhizobiales, Burkholderiales, Pseudomonadales, and Enterobacterales, which were the dominant bacterial orders for all *M. onukii* populations tested. A previous study showed that the order Enterobacterales is involved in essential amino acid biosynthesis in the tephritid fruit fly *Bactrocera dorsalis* (25). The Tax4Fun2 analysis results also showed that bacterial communities in *M. onukii* were involved in xenobiotics metabolism. For *M. onukii*, the sources of xenobiotic compounds ingested include plant

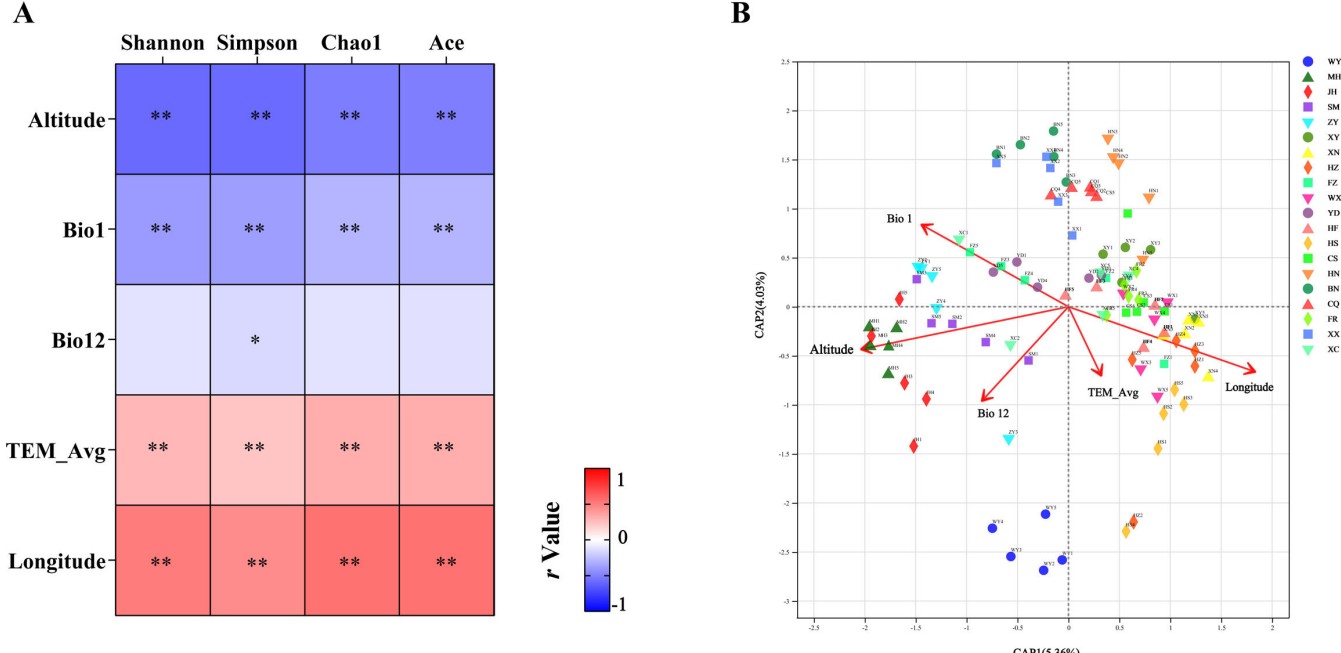

**FIG 8** Relationships between bacterial diversity and the main drivers. Drivers of bacterial alpha diversity across different environmental factors as determined by Spearman correlations (A); significance levels are denoted as follows: *$P < 0.05$; **$P < 0.01$; CAP showing environmental factors that influenced bacterial assembly (B). Bio1, annual mean temperatures; Bio12, annual mean precipitation; TEM_avg, the mean temperature for 30 days in total before and after the sampling date.

secondary compounds (e.g., caffeine, tea polyphenols, and tea saponin) and insecticide (e.g., afidopyropen, imidacloprid, and lufenuron) (26, 27). This indicates that the gut bacterial communities of *M. onukii* may have the ability to metabolize these compounds and contribute to their host's environmental adaptability and should therefore be further studied.

## Specific environmental factors affecting the bacterial communities of *M. onukii*

Naturally occurring populations of *M. onukii* may experience different selection pressures resulting from various pest management practices, natural enemies (e.g., pathogens, predators, and parasitoids), and environmental conditions influencing the frequency of their associated bacteria. The microbiomes of insects are highly susceptible to changes in their surrounding environment, including season, photoperiod, temperature, food, altitude, geography, and social interactions (28–32). The results of this study showed that the bacterial community structure of *M. onukii* collected from different regions differed significantly, and the host microbial community structure was shaped by environmental factors, including altitude, Bio1, Bio12, TEM_Avg, and longitude.

Based on our variance inflation factor (VIF) screening and CCA sorting, we suggest that altitude, temperature (Bio1, TEM_Avg), precipitation (Bio12), and longitude may affect microbial communities. In several related studies, altitude has been found that may affect host's microbial diversity (18, 19, 33). In this study, altitude showed significant correlation with the alpha diversity of the associated bacteria of *M. onukii* and contributed the most toward explaining variation in bacterial community structure. Further analysis showed that altitude was significantly and positively correlated with three endosymbionts (*Wolbachia*, *Asaia,* and *Rickettsia*). In particular, the bacterial communities of hemipterans such as aphids and planthoppers are more susceptible to the effects of endophytic bacteria (34, 35). Recently, several studies found that *Wolbachia* infections reduced *Sogatella furcifera* bacterial diversity and changed their communities (22, 36). Here, we speculate that altitude is a key environmental factor that has shaped the

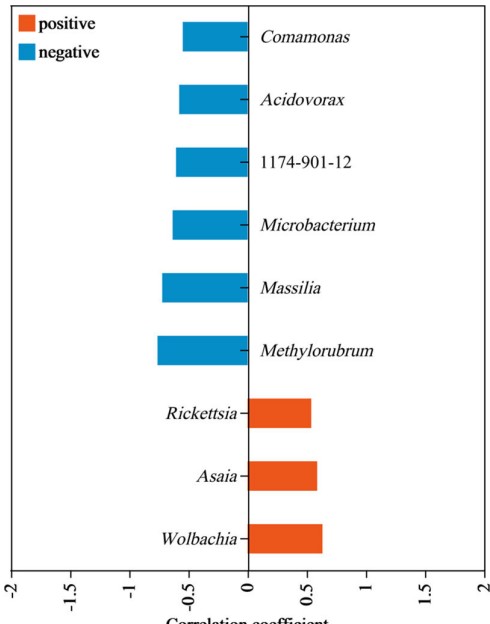

**FIG 9** Correlation of altitude to relative abundance of bacterial genera score by Spearman analysis. Positive, positive correlation; negative, negative correlation; $0.4 < r < 1$ representative significant positive correlation, and the larger the value, the stronger the correlation; $-1 < r < -0.4$ representative significant negative correlation, and the smaller the value, the stronger the correlation.

microbial communities of *M. onukii* by affecting the relative abundance of endosymbionts, especially *Wolbachia*.

To date, many effects of precipitation on insect microbiome have been reported. For example, precipitation can influence the incidence of *Spiroplasma*, a facultative endosymbiont that can manipulate host production (37). As *M. onukii* is polyphagous, any effects of longitude and precipitation may reflect effects of these variables on vegetation and food resources for *M. onukii*, which could alter the physiology and metabolism of *M. onukii* and in turn influence their microbial communities. Hence, different bacteria present in different environments could contribute to variation in microbial communities.

In summary, factors that shape the bacterial community of *M. onukii* include altitude, temperature, longitude, and rainfall. These factors collectively influence the environmental conditions, resource availability, and physiological processes that affect the growth and composition of bacterial populations within *M. onukii*. By considering the combined effects of these factors, we can gain a better understanding of the complex interactions between environmental factors and the bacterial community of *M. onukii*.

## Regional differences in the fungal communities of *M. onukii*

The fungal component of insects' microbiota is frequently overlooked, possibly because fungi are often associated with insects feeding on wood or detritus. Like insects' bacterial symbionts, their fungal symbionts also play a role in growth, development, and reproduction of sap-feeding insects (38). For example, entomopathogenic yeast-like symbionts play an important role in brown planthopper *Laodelphax striatellus* growth, reproduction, and nutrition (39). However, there have been no reports on the fungal communities associated with *M. onukii*. Here, we investigated the fungal community composition of *M. onukii* for the first time via the ITS region amplicon sequencing method. We concluded that *M. onukii* harbor very diverse fungal communities, which are influenced by host plant and habitat.

Unlike bacteria, fungal communities are more susceptible to environmental factors such as different diets and habitat. Previous research has suggested that the fungal portion of the house fly *Musca domestica* microbiota is highly dependent on the influx from the environment and less restricted by the physiology of the insects (40). Our collective sequencing evidence revealed that *M. onukii* harbor a variety of plant pathogenic fungi, such as unidentified_*Pleosporales*_sp., *Fusarium*, *Alternaria,* and *Gibberella*. These fungi may have come from the surrounding soil or other plants. Furthermore, at the genus level, *Cladosporium* was the dominant genus in all test samples. In our previous study, we found that *Cladosporium* was a common endosymbiotic fungi in tea leaves (41). Therefore, this could be one of the reasons why the diversity analysis results indicate no significant differences in the fungal communities of *M. onukii* collected from different regions. Furthermore, we conclude that the fungi of the *M. onukii* microbial communities mainly came from the endophytic fungi of the tea plants and the fungi in the surrounding environment. The FUNGuild analysis results confirmed our view. Based on this, we speculate that *M. onukii* may have the potential to transmit plant pathogens among different hosts.

## Conclusion

Together, our results show that *M. onukii* carry a complex and variable microbiota, which is influenced by geographical (i.e., altitude) as well as climate-related factors. Additionally, altitude as a key environmental factor may have shaped microbial communities of *M. onukii* by affecting the relative abundance of endosymbionts, especially *Wolbachia*. However, this study has many limitations. Our methods may have resulted in missing taxa. The great diversity of symbionts does not necessarily mean their greater absolute abundance. More detailed characterizations and functional analyses of microbes in *M. onukii* should be investigated via specifically designed experiments. Additionally, the mechanisms responsible for symbiont-host interactions should be explored.

## MATERIALS AND METHODS

### Sample collection

*Matsumurasca onukii* individuals were collected from tea plants at 20 distinct geographic locations, including 10 hilly area tea plantations (Hilly), and 10 mountainous region tea plantations (Mountain), in mainland China (covering all major tea-producing regions) during summer and autumn (May to October) of 2021 (Table S1; Fig. S1). To avoid host plant influence, all *M. onukii* were collected from tea plants in large-scale tea plantations with an area of more than 10 hectares. To avoid potential sampling bias arising from the collection of multiple-related individuals, we used the five-point sampling method (1,600 m$^2$) to collect *M. onukii* in each location. We randomly collected approximately 200–300 individuals at each location. *Matsumurasca onukii* were directly placed in sterile 50 mL Falcon tubes containing 100% ethanol and stored at −20℃.

The annual mean temperatures (Bio1) and annual mean precipitation (Bio12) of the 20 locations were obtained from DIVA-GIS v7.5.0 (https://www.diva-gis.org), which is a geographic information system for the analysis of species distribution data. The mean temperature for 30 days in total before and after the sampling date (TEM_Avg) and mean atmospheric pressure for 30 days in total before and after the sampling date (PRS_Avg) were obtained from the China Meteorological Administration (http://www.cma.gov.cn/).

### DNA extraction and species identification

Prior to DNA extraction, each *M. onukii* sample was surface sterilized by dipping in 75% ethanol for 5 min and then rinsing three times with sterile water for 15 s each time. Total genomic DNA was extracted with a DNeasy Blood and Tissue Kit (Qiagen, Hilden,

Germany) according to the manufacturer's protocols. The quality of the extracted DNA was assessed using electrophoresis on a 2% (wt/vol) agarose gel.

To ensure that the collected samples were *M. onukii*, adult females were randomly selected from each population for the mitochondrial *COI* gene (COIF: 5′-GGTCAACAAATCATAAAGATATTG-3′ and COIR: 5′-TAAACTTCAGGGTGACCAAAAAAT-3′) and 16sRNA (LR_J_12887: 5′-CCGGTYTGAACTCARATCAWGT-3′ and LR_N_13398: 5′-CTGTTTAWCAAAAACATTTC-3′) amplifications and sequencing according to Fu et al. (11), with slight modifications. The PCR reactions were carried out on an ABI Veriti thermocycler in 25 µL total volumes containing 8.5 µL of ddH$_2$O, 12.5 µL of Premix Taq (LA Taq Version 2.0, TaKaRa, Dalian, China), 2 µL of DNA template, and 1 µL of each primer (10 µM). The PCR conditions were as follows: initial denaturation at 94℃ for 3 min, followed by 30 cycles at 94℃ for 30 s, 50℃ for 1 min, 72℃ for 1 min, and a final extension at 72℃ for 10 min with a 4℃ hold. The quality of the PCR products was assessed using electrophoresis on a 2% (wt/vol) agarose. All the products were sequenced by Sangon Biotech (Shanghai, China).

The phylogenetic trees were constructed using the *COI* and 16S rRNA single gene and the maximum likelihood approach, implemented in MEGA v11.0 (Fig. S2). The nucleotide substitution model was determined via the Bayesian information criterion value calculated using MEGA v11.0 software. The substitution models were both Tamura 3-parameter (T92) with 1,000 bootstrap replicates.

## 16S rRNA and ITS region amplicon sequencing

The symbiotic bacterial and fungal communities of *M. onukii* were assessed using high-throughput amplicon sequencing of the 16S rRNA gene and the ITS region, respectively. A total of 100 *M. onukii* samples, including 50 Hilly and 50 Mountain samples, were used for 16S rRNA gene amplicon sequencing; five adults and five nymphs were pooled to provide a biological replicate, and five biological replicates were established per location. Then, for each location, we randomly chose one biological replicate, including nine Hilly and nine Mountain samples, for ITS1 region amplicon sequencing.

Genomic DNA was diluted to 1 ng/µL for PCR amplification. The V3-V4 region of the bacterial 16S rRNA gene was amplified using the primer set 338F (5′-ACTCCTAC GGGAGGCAGCAG-3′) and 806R (5′-GGACTACHVGGGTWTCTAAT-3′), and the fungal ITS region was amplified using the primer set ITS1-1F-F (5′-CTTGGTCATTTAGAGGAAGTAA-3′) and ITS1-1F-R (5′-GCTGCGTTCTTCATCGATGC-3′). All PCR reactions consisted of 15 µL of Phusion High-Fidelity PCR Master Mix with GC Buffer (New England Biolabs, Beverly, MA, United States), 2 µM of forward and reverse primers, and 10 ng of template DNA. The thermal cycling conditions were as follows: 98℃ for 1 min, followed by 30 cycles at 98℃ for 10 s, 50℃ for 30 s, 72℃ for 30 s, and a final extension at 72℃ for 5 min with a 4℃ hold. After PCR amplification, the samples were sequenced on the Illumina NovaSeq 6000 platform (Illumina, San Diego, CA, United States), and 250-bp paired-end reads were generated.

## Microbial community analysis

The Illumina NovaSeq sequencing of the bacterial 16S rRNA and the fungal ITS amplicon from *M. onukii* individuals yielded 8,650,259 and 1,538,327 raw reads in total, respectively. After quality filtering and read merging, a total of 7,704,410 high-quality sequences for bacteria and 1,446,785 high-quality sequences for fungi remained. The sequences were analyzed using the QIIME (v1.9.1) software package and used to compare the relative abundance of bacterial/fungal taxa (42). Sequence analysis was performed using UPARSE software (v7.0.1001). Sequences with ≥97% similarity were assigned to the same operational taxonomic units (43). A representative sequence for each OTU was screened for further annotation. For each representative sequence, the Silva Database was used based on the Mothur algorithm to annotate taxonomic

information (44, 45). Abundance values of OTUs were normalized using a standard value for the sample with the fewest sequences. Subsequent analyses of alpha diversity were performed using this normalized output.

To avoid the potential influence of primary symbiotic bacteria, we removed *Candidatus* Sulcia muelleri for subsequent analysis. After quality control and rarefaction cutoffs, 100 bacterial samples containing 6,512 bacterial OTUs and 18 fungal samples containing 1,398 fungal OTUs were left for subsequent analyses. The samples harbored a mean of 310.2 ± 124.3 bacterial OTUs (mean ± SE). A mean of 497.7 ± 163.5 fungal OTUs was identified from all samples. These OTUs were classified into 11 fungal phyla, 46 bacterial phyla, 466 fungal genera, and 1,112 bacterial genera. The alpha and beta diversity indexes were calculated using QIIME (v1.9.1) and displayed with R software (v2.15.3). Alpha diversity analyses were performed using observed OTUs for microbial richness and Shannon indexes for microbial species diversity (46). The diversity index values were compared using the Kruskal-Wallis test and *P*-values adjusted with the false discovery rate method. Rarefaction curves were used to verify the quality and depth of the sampling (Fig. S3). Beta diversity comparisons were performed using the unweighted UniFrac method for presence/absence metrics and the weighted UniFrac method for relative abundance metrics by non-metric multidimensional scaling. The distance matrices were further analyzed using an adonis analysis (also known as permutational multivariate analysis of variance) with 999 permutations to compare the differences in microbial communities between Hilly and Mountain samples. LEfSe analysis was conducted to reveal the significant ranking of the abundant modules.

For correlation analyses, alpha diversity indices and relative abundances of dominant bacteria (genus level) were processed in IBM SPSS Statistics v26 using correlation analysis (Spearman correlation and Pearson correlation). Environmental factors were divided into two categories, which included geographical and climatic factors; VIF coefficients were used to screen meaningful environmental factors. VIF analysis results showed that PRS_Avg and latitude were two invalid environmental factors (VIF >10). Therefore, the subsequent correlation analysis did not include these two factors. CCA was used to calculate the relationships between environmental factors and *M. onukii* microbiota communities. To test the significance and importance of the environmental variables for beta diversity, we used a distance-based linear model and forward selection procedure based on the Bray-Curtis distance matrix by estimating the proportion of variance explained ($R^2$). These results were displayed by CAP analysis. The microbiota function of *M. onukii* was predicted by Tax4Fun2 (bacteria) and FUNGuild (fungi) (47–49).

## ACKNOWLEDGMENTS

Y.Y., C.G., and Y.Z. contributed to conception and design of the study. Y.Z., wrote the manuscript. Y.Y. and C.G. led the writing (review and editing). S.L., H.Z., T.G., R.J., and D.Z. collected the samples and information. X.H., Y.Z., J.S., and S.L. performed the experiments. Y.Z. and Y.-L.Z. revised and proofread the manuscript. All authors approved the final version for submission.

This work was supported by the National Key Research and Development Program of China (No.2021YFD1601105); National Natural Science Foundation of China (No.32172635); Joint Funds of National Natural Science Foundation of China (No.U21A20232); Research team on accumulation and control mechanism of risk substances in tea, Anhui Excellent Scientific Research and Innovation team (2022AH010055); Post Experts of the Tea Industry (AHCYTSTX-11).

## AUTHOR AFFILIATIONS

[1]State Key Laboratory of Tea Plant Biology and Utilization, Anhui Agricultural University, Hefei, Anhui, China
[2]College of Horticulture, Northwest A&F University, Yangling, Shaanxi, China
[3]College of Life Science, Anhui Agricultural University, Hefei, Anhui, China

## AUTHOR ORCIDs

Yong Zhang ⓘ http://orcid.org/0000-0002-7392-9406
Chun-mei Gong ⓘ http://orcid.org/0000-0002-1164-5440

## FUNDING

| Funder | Grant(s) | Author(s) |
| --- | --- | --- |
| MOST | National Key Research and Development Program of China (NKPs) | No.2021YFD1601105 | Yun-qiu Yang |
| MOST | National Natural Science Foundation of China (NSFC) | No.32172635 | Chun-mei Gong |

## AUTHOR CONTRIBUTIONS

Song Liu, Investigation, Methodology | Xue-yu Huang, Methodology | Hua-bin Zi, Software | Tian Gao, Methodology | Rui-jie Ji, Methodology | Juan Sheng, Methodology | Dian Zhi, Software | Ying-lao Zhang, Writing – review and editing | Chun-mei Gong, Conceptualization, Funding acquisition, Supervision, Writing – review and editing | Yun-qiu Yang, Conceptualization, Funding acquisition, Project administration, Supervision, Visualization, Writing – review and editing.

## DATA AVAILABILITY

The sequences of the COI and 16S rRNA genes of all *M. onukii* were deposited in GenBank (accession numbers: OP752247-OP752266, OP748894-OP748913). All amplicon sequences are available as SRA files at the National Center for Biotechnology Information Sequence Read Archive database (NCBI-SRA) under bioProject PRJNA720281.

## ADDITIONAL FILES

The following material is available online.

### Supplemental Material

**Supplemental Figures (Spectrum01009-23-S0001.docx).** Figures S1 to S9.
**Table S1 (Spectrum01009-S0002.xlsx).** Summary of collection details.

### Open Peer Review

**PEER REVIEW HISTORY (review-history.pdf).** An accounting of the reviewer comments and feedback.

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
