## [Reviewer comments · Microbiology Spectrum]

Microbiology Spectrum

Altitude as a key environmental factor shaping microbial communities of tea green leafhoppers (*Matsumurasca onukii*)

Yong Zhang, Song Liu, Yu Huang, Bin Zi, Tian Gao, Jie Ji, Juan Shen, Dian Zhi, Lao Zhang, Mei Gong, and Qiu Yang

Corresponding Author(s): Yong Zhang, Northwest A&F University College of Horticulture

Review Timeline:

Submission Date:	March 7, 2023
Editorial Decision:	June 6, 2023
Revision Received:	July 9, 2023
Accepted:	September 19, 2023

Editor: Wenli Chen

Reviewer(s): The reviewers have opted to remain anonymous.

Transaction Report:

DOI: <https://doi.org/10.1128/spectrum.01009-23>

June 6, 2023

Dr. Yong Zhang
Northwest A&F University College of Horticulture
Xinong road 22
Yangling
China

Re: Spectrum01009-23 (**Altitude as a key environmental factor shaping microbial communities of tea green leafhoppers (*Matsumurasca onukii*)**)

Dear Dr. Yong Zhang:

Thank you for submitting your manuscript to Microbiology Spectrum. It has been reviewed by the experts. Copies of the reviewers' comments are appended below for your consideration. I am not able to accept the current submission for publication. Nonetheless, I encourage you to consider the comments of the reviewers for improving your manuscript, and to submit the revised manuscript.

Link Not Available

Sincerely,

Wenli Chen

Journals Department
Reviewer comments:

Reviewer #1 (Comments for the Author):

Tea is a popular beverage in the world. As the main production area of tea, China has a long history in the cultivation and management of tea. *Matsumurasca onukii* is an important destructive pest in the process of tea cultivation, and effective control of it has become the basis for safe production of tea. The insect microbiome has an important impact on the environmental adaptability and physiological indicators of insects. Revealing the diversity and relationship between the microbiome of the *M. onukii* and environmental factors is the foundation for in-depth research on the functions of the *M. onukii* microbiome in the

future. Zhang et al. analyzed the diversity of the microbiome of *M. onukii* in major tea producing regions in China, and analyzed the relationship between its diversity and environmental factors. This is of great significance for enriching the research on the microbiome and its functions of *M. onukii*. The following suggestions are provided for the authors in order to improve the manuscript:

1. L66-69: References required;
2. L70: The above statement cannot reach this conclusion. Please describe it more rigorously;
3. The purpose of the manuscript in the introduction is only to analyze the microbiome diversity of *M. onukii* and its altitude significance which are too descriptive, and it is recommended to further refine the purpose and significance of the study;
4. Figure S6 is meaningful information, and it is recommended to put it in the maintext;
5. There are many abbreviations in the ms that are not easy to understand and need further clarification;
6. The diversity of fungi in the study does not seem to vary significantly between different altitude, especially the core flora, so explanations need to be made during the discussion;
7. Although altitude is an important environmental factor that affects microbial diversity in the study, other factors also play a role, and explanations should also be made during the discussion to avoid misleading the reader. At the same time, appropriate additions and changes should be made in the abstract;
8. L391 : Why only females? Because microbiome analysis did not distinguish between male and female;
9. L414-416: Why mix adults and nymphs? There are differences between them;
10. L417: Why are there only 9 samples? As there are 10 sampling points;
11. Figure legends are not detailed enough, additional information is needed.

Reviewer #2 (Comments for the Author):

Insects frequently harbor a variety of symbiotic microbiota, and the interactions between them may have important effects on their evolution. However, the factors shaping microbial communities in wild populations remain largely unknown. Here, a comprehensive study is carried on altitude as a key environmental factor shaping microbial communities of tea green leafhopper. The experimental design is reasonable, and the findings are fascinating. More advanced metagenomes can be looked at for future work, and more novel data may be presented.

Staff Comments:

Preparing Revision Guidelines

Please return the manuscript within 60 days; if you cannot complete the modification within this time period, please contact me. If

you do not wish to modify the manuscript and prefer to submit it to another journal, please notify me of your decision immediately so that the manuscript may be formally withdrawn from consideration by Microbiology Spectrum.

Insects frequently harbor a variety of symbiotic microbiota, and the interactions between them may have important effects on their evolution. However, the factors shaping microbial communities in wild populations remain largely unknown. Here, a comprehensive study is carried on altitude as a key environmental factor shaping microbial communities of tea green leafhopper. The experimental design is reasonable, and the findings are fascinating. Before the paper is accepted, here are some minor comments:

Many studies have shown that bacteria play an important role in symbiotic bacteria for stinging and sucking mouthparts. I saw that your Introduction also mainly discussed the influence of bacteria on some insects, but your experiment also designed the high-throughput sequencing of ITS. Can you explain why this experiment was designed?

Line367-370: How to distinguish mountain tea plantations and hilly tea plantations? What is the sampling point selection criteria?

Line 391-392: What is the purpose of amplifying mitochondrial *COI* gene and 16sRNA gene ?
What is the niche of *Matsumurasca onukii*.

Results

Line168-169: The author emphasizes the ectosymbionts and endosymbionts. What is the reason for separating them for analysis.

References

In the references, there are some errors in abbreviations and italics of some journal names, such as 10, 11, and some Latin names are not in italics, such as 38, please correct them.

List of Responses

Dear Prof Chen and Reviewers

Thank you for your letter and for the reviewers' comments concerning our manuscript entitled "**Altitude as a key environmental factor shaping microbial communities of tea green leafhoppers (*Matsumurasca onukii*) (Spectrum01009-23)**". "Under Peer Review" on 2023-03-10 and "Manuscript in Revision" on 2023-06-06. Those comments are all valuable and very helpful for revising and improving our paper, as well as the important guiding significance to our researches. We have studied comments carefully and have made correction which we hope meet with approval. The main corrections in the paper and the responds to the reviewer's comments are as flowing:

Reviewer comments:

Reviewer #1 (Comments to the Author):

Dear Reviewer:

Thank you very much for your recognition of our work, and put forward many valuable opinions. Based on your comment and request, we have made extensive modification on the original manuscript. Here, we attached revised manuscript in the formats of both PDF and MS word, for your approval. A revised manuscript with the correction sections red marked was attached as the supplemental material and for easy check purpose.

Here below is our description on revision:

Introduction**Q1. L66-69: References required**

Responds: Thank you for your reminding. This example is a detailed description of the literature cited in the previous sentence (i.e., “**For example, microbiome infection frequencies determine the geographic distribution of the chestnut weevil, *Curculio sikkimensis*”**”).

Q2. L70: The above statement cannot reach this conclusion. Please describe it more rigorously;

Responds: We have deleted this sentence. At the same time, the paragraph has been slightly modified. Please check in on Line 71-73

Q3. The purpose of the manuscript in the introduction is only to analyze the microbiome diversity of *M. onukii* and its altitude significance which are too descriptive, and it is recommended to further refine the purpose and significance of the study;

Responds: Thank you very much for your suggestion, we modify it in the last paragraph of the Introduction part. Please check it on Line 115-120

“Not only that, but the factors influencing microbial communities in wild populations, including environmental factors and interactions among microbial species, are still poorly understood. Given the tea green leafhopper’s broad geographical distribution and high adaptability, it serves as a suitable model for studying the impact of ecological drivers on microbiomes.”

Results

Q4. Figure S6 is meaningful information, and it is recommended to put it in the maintext;

Responds: Thank you for your reminding. We have put Figure S6 in the maintext of the manuscript. Please check it on Line 173.

Q5. There are many abbreviations in the ms that are not easy to understand and need further clarification;

Responds: I am sincerely sorry for the writing mistakes that have caused you trouble. During the writing process, I inadvertently placed the "MATERIALS AND METHODS" section after the "INTRODUCTION", leading to some abbreviations being explained in the wrong section. However, according to the submission format for the manuscript, the "MATERIALS AND METHODS" section should be placed after the "CONCLUSION". As a result, some abbreviations in the "RESULTS" and "DISCUSSION" sections may not be easily understood. We have thoroughly reviewed and modified the abbreviations in this manuscript to address this issue."

Q11. Figure legends are not detailed enough, additional information is needed.

Responds: Thank you very much for your suggestion, we have carefully reviewed and rewritten Figure legends. Please check it on Line 681-734

Discussion

Q6. The diversity of fungi in the study does not seem to vary

significantly between different altitude, especially the core flora, so explanations need to be made during the discussion;

Responds: We have revised it. Please check it on Line Line 363-365.

Altitude alone may not be the primary factor influencing fungal diversity in tea green leafhoppers. Other factors such as geographic location, host specificity, and ecological interactions could have a more substantial impact on shaping the fungal composition. These additional factors might outweigh the influence of altitude, resulting in a lack of significant differences observed in the fungal communities across different regions.

Q7. Although altitude is an important environmental factor that affects microbial diversity in the study, other factors also play a role, and explanations should also be made during the discussion to avoid misleading the reader. At the same time, appropriate additions and changes should be made in the abstract;

Responds: We have revised it. Please check it on Line 336-342.

Materials and methods

Q8. L391: Why only females? Because microbiome analysis did not distinguish between male and female;

Responds: The amplification of the COI fragment of the mtDNA and 16S rRNA from the female tea green leafhopper was conducted to confirm the identity of the insect samples used in this experiment as *Matsumurasca onukii*. This step helped ensure that the collected samples were indeed the target species.

Subsequently, for the microbiome analysis, a mixed sample comprising both nymphs and adults (including both males and females) of the tea green leafhopper was used. This approach allowed for a comprehensive assessment of the microbial community associated with the tea green leafhopper across different life stages and

genders.

Q9. L414-416: Why mix adults and nymphs? There are differences between them;

Responds: This is an important issue to consider. The decision to mix nymphs and adults together in our study was based on the following reasons:

1) Neither nymphs nor adults alone can fully represent the population of leafhoppers. By mixing both nymphs and adults together, we aimed to obtain a more comprehensive representation of the leafhopper population. Additionally, processing all samples in the same manner ensures consistency and comparability of the data.

2) The tea green leafhopper undergoes incomplete metamorphosis, and the nymphs and adults share similar morphology and feeding patterns. As a result, we anticipated that there would be minimal differences in the microbiome between the two groups. We also investigated the bacterial component of the microbiome carried by tea green leafhoppers at different developmental stages, and the corresponding results are presented in **Table 1, Figure 1 and Figure 2**

Table 1 Analysis of alpha diversity of gut bacteria in tea green leafhoppers at different developmental stages

Index	Low-instar (L)	High-instar (H)	Adults (A)	P value
Observed_ASVs	313.66 ± 2.12	277.28 ± 48.90	246.56 ± 29.65	> 0.05 ^{ns}
Dominance	0.016 ± 0.006	0.013 ± 0.005	0.018 ± 0.009	> 0.05 ^{ns}
Shannon	6.94 ± 0.24	7.04 ± 0.21	6.71 ± 0.64	> 0.05 ^{ns}
Simpson	0.98 ± 0.01	0.99 ± 0.01	0.99 ± 0.01	> 0.05 ^{ns}

The diversity index values were compared with Kruskal-Wallis test results after false discovery (FDR) correction; $P > 0.05$ represent no significantly difference

Figure 1 Non-metric Multidimensional Scaling (NMSD) analysis based on Weighted_Unifrac and

Unweighted_Unifrac distance (D) calculation the Beta diversity of the gut bacterial community

Significant differences were detected by Adonis test; $P > 0.05$ represent no significantly difference

Figure 3 Composition of gut bacteria in the tea green leafhopper at different developmental stages at six bacterial levels: phyla, class, order, family, genus and species

10. L417: Why are there only 9 samples? As there are 10 sampling points;

Responds: I apologize for the confusion in the manuscript. Thank you for providing further clarification. The reasons for selecting a random group of samples from the CS and FR sampling points, as well as from the CQ and BN sampling points, are as follows:

CS and FR sampling points: These two points were in close proximity, and based on preliminary analysis, there were no significant differences observed in the bacterial composition between them. Therefore, to ensure representative sampling without redundancy, a random selection was made from these points.

CQ and BN sampling points: Similarly, these two points showed similarities in bacterial composition, and no significant differences were identified during preliminary analysis. Thus, a random sampling approach was employed to capture representative samples without duplication.

Reviewer #2 (Comments to the Author):

Dear Reviewer:

I am very grateful to your comments for the manuscript. According with your advice, we amended the relevant part in manuscript. Here, we attached revised manuscript in the formats of both PDF an MS word, for your approval. A revised manuscript with the correction sections red marked was attached as the supplemental material and for easy check purpose.

Here below is our description on revision:

Major revised:

Many studies have shown that bacteria play an important role in symbiotic bacteria for stinging and sucking mouthparts. I saw that your Introduction also mainly discussed the influence of bacteria on some insects, but your experiment also designed the high-throughput sequencing of ITS. Can you explain why this experiment was designed?

Responds: Firstly, I sincerely appreciate your recognition of our work. I would also like to express my gratitude for pointing out the shortcomings in our manuscript and providing valuable suggestions for revision. The decision to incorporate high-throughput sequencing of ITS was based on the following reasons:

1) Similar to bacteria, symbiotic fungi are an essential component of the leaf-sap insect microbiome, influencing the growth, development, and reproduction of sap-feeding insects. However, the specific composition and diversity of fungal communities associated with tea green leafhoppers remain largely unknown.

2) Leafhoppers exhibit complex feeding habits, extracting sap from a variety of plant species. Consequently, many leafhoppers have the potential to transmit plant viruses. Additionally, numerous fungi can cause significant damage to plants, resulting in substantial economic losses. Examples include *Fusarium graminearum*, *Fusarium oxysporum*, *Ustilago maydis*, and *Gibberella zeae*. It

remains unclear whether tea green leafhoppers can serve as carriers of these pathogenic fungi. Therefore, we included relevant content regarding the fungal composition of leafhoppers in our study. The aim is to investigate whether wild populations of leafhoppers harbor a significant presence of plant pathogenic fungi.

Q1: Line 367-370: How to distinguish mountain tea plantations and hilly tea plantations? What is the sampling point selection criteria?

Responds: I apologize for not explaining the issue clearly.

Elevation: Mountain tea plantations are typically located at higher elevations compared to hilly tea plantations. Mountain tea is often grown at altitudes above 400 meters, whereas hilly tea plantations are found at lower elevations.

Slope and Terrain: Mountain tea plantations are usually situated on steeper slopes with rugged terrains. These areas often have more challenging topography compared to hilly tea plantations, which are typically located on gentler slopes.

Q2: Line 391-392: What is the purpose of amplifying mitochondrial COI gene and 16S rRNA gene? What is the niche of *Matsumurasca onukii*.

Responds: The leafhopper pests found in tea plantations encompass not only tea green leafhoppers but also other species such as *Empoasca vitis*, *Empoasca limbifera* Matsumura and *Balclutha graminea* Merino. Although tea green leafhoppers dominant more than 90% of the ecological niche in tea plantations, it is important to note that other leafhopper pests are present as well.

Furthermore, due to the similarities in morphology among different leafhopper species, it becomes challenging to differentiate them visually. Therefore, molecular methods are necessary for accurate identification. In order to ensure that the insect samples used in the experiment were tea green leafhoppers, we amplified two fragments of the insect host's mitochondrial DNA sequence (COI and 16S rRNA).

Results

Line168-169: The author emphasizes the ectosymbionts and endosymbionts. What is the reason for separating them for analysis.

Responds: Ectosymbionts and endosymbionts are two types of symbiotic relationships that organisms can form with other species.

Localization: Endosymbionts reside within the body of the insect host, often in specialized organs or cells such as the gut or bacteriocytes. Ectosymbiotic, on the other hand, colonize the external surfaces of the insect, such as the gut or antennae.

Direct Impact: Endosymbiotic bacteria have a more direct impact on the physiology and development of the insect host. They can influence various aspects of the insect's biology, including reproduction, metabolism, and immunity. Ectosymbiotic bacteria, while still interacting with the host, may have more limited effects and primarily provide certain benefits like protection against predators or pathogens.

Nutritional Contributions: Endosymbiotic bacteria often play a crucial role in providing essential nutrients to the insect host. For example, they can synthesize vitamins, amino acids, or other compounds that are deficient or absent in the insect's diet. Ectosymbiotic bacteria generally do not have as prominent a role in nutrient provisioning.

Therefore, we separate the analysis of endosymbiotic and ectosymbiotic because they have distinct roles and effects on the insect host. By studying them separately, we can gain a deeper understanding of the specific contributions and mechanisms of each type of symbiont in shaping the biology, physiology, and ecology of the insect. This separation allows us to appreciate the unique interactions and adaptations that have evolved between insects and their symbiotic partners in different ecological contexts.

References

In the references, there are some errors in abbreviations and Italics of some journal names, such as 10, 11, and some Latin names are not in italics, such as 38, please correct them.

Responds: Thank you very much for pointing out the errors. We have diligently corrected the mistakes in the reference. Please check.

September 19, 2023

Dr. Yong Zhang
Northwest A&F University College of Horticulture
Xinong road 22
Yangling
China

Re: Spectrum01009-23R1 (**Altitude as a key environmental factor shaping microbial communities of tea green leafhoppers (*Matsumurasca onukii*)**)

Dear Dr. Yong Zhang:

Thank you very much for submitting your manuscript to Microbiology Spectrum. It has been reviewed by the experts. I am glad to inform you that your manuscript has been accepted, and I am forwarding it to the ASM Journals Department for publication. You will be notified when your proofs are ready to be viewed.

Sincerely,

Wenli Chen
Editor, Microbiology Spectrum
